# Preference-based Reinforcement Learning with Finite-Time Guarantees

**Yichong Xu**[1]*, **Ruosong Wang**[2], **Lin F. Yang**[3], **Aarti Singh**[2], **Artur Dubrawski**[2]
[1]Microsoft
[2]Carnegie Mellon University
[3]University of California, Los Angles
yicxu@microsoft.com, {ruosongw,aarti,awd}@cs.cmu.edu, linyang@ee.ucla.edu

## Abstract

Preference-based Reinforcement Learning (PbRL) replaces reward values in traditional reinforcement learning by preferences to better elicit human opinion on the target objective, especially when numerical reward values are hard to design or interpret. Despite promising results in applications, the theoretical understanding of PbRL is still in its infancy. In this paper, we present the first finite-time analysis for general PbRL problems. We first show that a unique optimal policy may not exist if preferences over trajectories are deterministic for PbRL. If preferences are stochastic, and the preference probability relates to the hidden reward values, we present algorithms for PbRL, both with and without a simulator, that are able to identify the best policy up to accuracy $\varepsilon$ with high probability. Our method explores the state space by navigating to under-explored states, and solves PbRL using a combination of dueling bandits and policy search. Experiments show the efficacy of our method when it is applied to real-world problems.

## 1 Introduction

In reinforcement learning (RL), an agent typically interacts with an unknown environment to maximize the cumulative reward. It is often assumed that the agent has access to numerical reward values. However, in practice, reward functions might not be readily available or hard to design, and hand-crafted rewards might lead to undesired behaviors, like reward hacking [8, 1]. On the other hand, preference feedback is often straightforward to specify in many RL applications, especially those involving human evaluations. Such preferences help shape the reward function and avoid unexpected behaviors. Preference-based Reinforcement Learning (PbRL, [28]) is a framework to solve RL using preferences, and has been widely applied in multiple areas including robot teaching [20, 19, 11], game playing [29, 31], and in clinical trials [36].

Despite its wide applicability, the theoretical understanding of PbRL is largely open. To the best of our knowledge, the only prior work with a provable theoretical guarantee is the recent work by Novoseller et al. [25]. They proposed the Double Posterior Sampling (DPS) method, which uses Bayesian linear regression to derive posteriors on reward values and transition distribution. Combining with Thompson sampling, DPS has an asymptotic regret rate sublinear in $T$ (number of time steps). However, this rate is based on the *asymptotic* convergence of the estimates of reward and transition function, whose complexity could be exponential in the time horizon $H$. Also, the Thompson sampling method in [25] can be very time-consuming, making the algorithm applicable only to MDPs with a few states. To fill this gap, we naturally ask the following question:

**Is it possible to derive efficient algorithms for PbRL with finite-time guarantees?**

While traditional value-based RL has been studied extensively, including recently [6, 35, 22], PbRL is much harder to solve than value-based RL. Most efficient algorithms for value-based RL utilize the value function and the Bellman update, both of which are unavailable in PbRL: the reward values are hidden and unidentifiable up to shifts in rewards. Even in simple tabular settings, we cannot obtain unbiased estimate of the Q values since any offset in reward function results in the same preferences. Therefore traditional RL algorithms (such as Q learning or value iteration) are generally not applicable to PbRL.

**Our Contributions.** We give an affirmative answer to our main question above, under general assumptions on the preference distribution.

- We study conditions under which PbRL can recover the optimal policy for an MDP. In particular, we show that when comparisons between trajectories are noiseless, there exists an MDP such that preferences between trajectories are not transitive; i.e., there is no unique optimal policy (Proposition 1). Hence, we base our method and analysis on a general assumption on preferences between trajectories, which is a generalization of the linear link function assumption in [25].

- We develop provably efficient algorithms to find $\varepsilon$-optimal policies for PbRL, with or without a simulator. Our method is based on a synthetic reward function similar to recent literature on RL with rich observations [14, 24] and reward-free RL [23]. We combine this reward-free exploration and dueling bandit algorithms to perform policy search. To the best of our knowledge, this is the first PbRL algorithm with finite-time theoretical guarantees. Our method is general enough to incorporate many previous value-based RL algorithms and dueling bandit methods as a subroutine.

- We test our algorithm against previous baselines in simulated environments. Our results show that our algorithm can beat previous baselines, while being very simple to implement.

**Related Work.** We refer readers to [28] for a complete overview of PbRL. In the PbRL framework, there are several ways that one can obtain preferences: We can obtain preferences on i) trajectories, where the labeler tells which of two trajectories is more rewarding; ii) actions, where for a fixed state $s$, the labeler tells which action gives a better action-value function; or iii) states, where similarly, the labeler tells which state gives a better value function. In this paper, we mainly study preferences over trajectories, which is also the most prevalent PbRL scenario in literature. Our method is potentially applicable to other forms of preferences as well.

PbRL is relevant to several settings in Multi-Armed Bandits. Dueling bandits [33, 10] is essentially the one-state version of PbRL, and has been extensively studied in the literature [34, 17, 16, 32]. However, PbRL is significantly harder because in PbRL the observation (preference) is based on the *sum* of rewards on a trajectory rather than individual reward values. For the same reason, PbRL is also close to combinatorial bandits with full-bandit feedback [12, 2, 26]. Although lower bounds for these bandit problems extends to PbRL, developing PbRL algorithms is significantly harder since we are not free to choose any combination of state-action pairs.

## 2 Problem Setup

**MDP Formulation.** Suppose a finite-time Markov Decision Process (MDP) $(\mathcal{S}, \mathcal{A}, H, r, p, s_0)$, where $\mathcal{S}$ is the state space, $\mathcal{A}$ is the action space, $H$ is the number of steps, $r : \mathcal{S} \times \mathcal{A} \to \mathbb{R}$ is the reward function[2], $p : \mathcal{S} \times \mathcal{A} \to \Delta(\mathcal{S})$ is the (random) state transition function, and $s_0$ is the starting state. For simplicity we assume $S \geq H$. We consider finite MDPs with $|\mathcal{S}| = S$, $|\mathcal{A}| = A$. We start an episode from the initial state $s_0$, and take actions to obtain a trajectory $\tau = \{(s_h, a_h)\}_{h=0}^{H-1}$, following a policy $\pi : \mathcal{S} \to \mathcal{A}$. We also slightly overload the notation to use $r(\tau) = \sum_{(s,a) \in \tau} r(s, a)$ to denote the total reward of $\tau$. For any policy $\pi$, let $\tau_h(\pi, s)$ be a (randomized) trajectory by executing $\pi$ starting at state $s$ from step $h$ to the end.

Following prior works [14, 24], we further assume that the state space $\mathcal{S}$ can be partitioned into $H$ disjoint sets $\mathcal{S} = \mathcal{S}_1 \cup \cdots \mathcal{S}_H$, where $\mathcal{S}_h$ denotes the set of possible states at step $h$. Let $\Pi : \{\pi : \mathcal{S} \to \mathcal{A}\}$ be the set of policies[3]. We use $\Pi^H$ to denote the set of non-stationary policies;

here a policy $\pi = (\pi_1, ..., \pi_H) \in \Pi^H$ executes policy $\pi_h$ in step $h$ for $h \in [H]$ [4]. Also, let $\pi_{h_1:h_2}$ be the restriction of policy $\pi \in \Pi^H$ to step $h_1, h_1 + 1, ..., h_2$. We use the value function $v_{h_0}^{\pi}(s) = \mathbb{E}[\sum_{h=h_0}^{H-1} r(s_h, \pi(s_h))|\pi, s_{h_0} = s]$ to denote the expected reward of policy $\pi$ starting at state $s$ in step $h_0$; for simplicity let $v^{\pi} = v_0^{\pi}$. Let $\pi^* = \arg\max_{\pi \in \Pi^H} v_0^{\pi}(s_0)$ denote the optimal (non-stationary) policy. We assume that $r(s, a) \in [0, 1]$ for every $(s, a) \in \mathcal{S} \times \mathcal{A}$, and that $v^*(s) = v^{\pi^*}(s) \in [0, 1]$ for every state $s$.

**Preferences on Trajectories.** In PbRL, the reward $r(s, a)$ is hidden and is not observable during the learning process, although we define the value function and optimal policy based on $r$. Instead, the learning agent can query to compare two trajectories $\tau$ and $\tau'$, and obtain a (randomized) preference $\tau \succ \tau'$ or $\tau' \succ \tau$, based on $r(\tau)$ and $r(\tau')$. We also assume that we can compare partial trajectories; we can also compare two partial trajectories $\tau = \{(s_h, a_h)\}_{h=h_0}^{H}$ and $\tau' = \{(s_h', a_h')\}_{h=h_0}^{H}$ for any $h_0 \in [H]$. Let $\phi(\tau, \tau') = \Pr[\tau \succ \tau'] - 1/2$ denote the preference between $\tau$ and $\tau'$.

**PAC learning and sample complexity.** We consider PAC-style algorithms, i.e., an algorithm needs to output a policy $\hat{\pi}$ such that $v^{\hat{\pi}}(s_0) - v^*(s_0) \leq \varepsilon$ with probability at least $1 - \delta$. In PbRL, comparisons are collected from human workers and the trajectories are obtained by interacting with the environment. So obtaining comparisons are often more expensive than exploring the state space; we therefore compute separately the sample complexity in terms of the number of total steps and number of comparisons. Formally, let $\text{SC}_s(\varepsilon, \delta)$ be the number of exploration steps needed to obtain an $\varepsilon$-optimal policy with probability $1 - \delta$, and $\text{SC}_p$ be the number of preferences (comparisons) needed in this process. We omit $\varepsilon, \delta$ from the sample complexities when the context is clear.

**Dueling Bandit Algorithms.** Our proposed algorithms uses a dueling bandit algorithm as a subroutine. To utilize preferences, our algorithms use a PAC dueling bandit algorithm to compare policies starting at the same state. Dueling Bandits [33] has been well studied in the literature. Examples of PAC dueling bandit algorithms include Beat the Mean [34], KNOCKOUT [17], and OPT-Maximize [16]. We formally define a dueling bandit algorithm below.

**Definition 1** (($\varepsilon, \delta$)-correct Dueling Bandit Algorithm). *Let $\varepsilon > 0, \delta > 0$. $\mathcal{M}$ is a ($\varepsilon, \delta$)-correct PAC dueling bandit algorithm if for any given set of arms $\mathcal{X}$ with $|\mathcal{X}| = K$, i) $\mathcal{M}$ runs for at most $\Psi(\varepsilon, \delta)\varepsilon^{-\alpha}$ steps, where $\Psi(\varepsilon, \delta) = poly(K, \log(1/\varepsilon), \log(1/\delta))$ and $\alpha \geq 1$; ii) in every step, $\mathcal{M}$ proposes two arms $a, a'$ to compare; iii) Upon completion, $\mathcal{M}$ returns an arm $\hat{a}$ such that $\Pr[\hat{a} \succ a] \geq 1/2 - \varepsilon$ for every arm $a \in \mathcal{X}$, with probability at least $1 - \delta$.*

One important feature of existing PAC dueling bandit algorithms is whether they require knowing $\varepsilon$ before they start - algorithms like KNOCKOUT and OPT-Maximize [17, 16] cannot start without knowledge of $\varepsilon$; Beat-the-Mean [34] does not need to know $\varepsilon$ to start, but can return an $\varepsilon$-optimal arm when given the correct budget. We write $\mathcal{M}(\mathcal{X}, \varepsilon, \delta)$ for an algorithm with access to arm set $\mathcal{X}$, accuracy $\varepsilon$ and success probability $1 - \delta$; we write $\mathcal{M}(\mathcal{X}, \delta)$ for a dueling bandit algorithm without using the accuracy $\varepsilon$.

**Results in the value-based RL and bandit setting.** In traditional RL where we can observe the reward value at every step, the minimax rate is largely still open [21] [5]. The upper bound in e.g., [6] translates to a step complexity of $O\left(\frac{H^3 SA}{\varepsilon^2}\right)$ to recover an $\varepsilon$-optimal policy, but due to scaling of the rewards the lower bound [13] translates to $\Omega\left(\frac{HSA}{\varepsilon^2}\right)$. Very recently, Wang et al. [27] show an upper bound of $O\left(\frac{HS^3 A^2}{\varepsilon^3}\right)$, showing that the optimal $H$ dependence might be linear. It is straightforward to show that the lower bound in [13] translates to a step complexity of $\Omega\left(\frac{HSA}{\varepsilon^2}\right)$ and a comparison complexity of $\Omega\left(\frac{SA}{\varepsilon^2}\right)$ for PbRL. Lastly, we mention that the lower bounds for combinatorial bandits with full-bandit feedback [12] can transform to a lower bound of the same scale for PbRL.

## 2.1 Preference Probablities

As in ranking and dueling bandits, a major question when using preferences is how to define the winner. One common assumption is the existence of a Condorcet winner: Suppose there exists an item (in our case, a policy) that beats all other items with probability greater than 1/2. However in PbRL, because we compare trajectories, preferences might not reflect the true relation between

policies. For example, assume that the comparisons are perfect, i.e., $\tau_1 \succ \tau_2$ if $r(\tau_1) > r(\tau_2)$ and vice versa. Now suppose policy $\pi_1$ has a reward of 1 with probability 0.1 and 0 otherwise, and $\pi_2$ has a reward of 0.01 all the time. Then a trajectory from $\pi_1$ is only preferred to a trajectory from $\pi_2$ with probability 0.1 if the comparisons give deterministic results on trajectory rewards. Extending this argument, we can show that non-transitive relations might exist between policies:

**Proposition 1.** *Slightly overloading the notation, for any $h \in [H]$ and $s_h \in \mathcal{S}_h$, let $\phi_s(\pi_1, \pi_2) = \Pr[\tau_h(\pi_1, s) \succ \tau_h(\pi_2, s)] - 1/2$ denote the preference between policies $\pi_1$ and $\pi_2$ when starting at $s_h$ in step $h$. Suppose comparisons are noiseless. There exists a MDP and policies $\pi_0, \pi_1, \pi_2$ such that for some state $s \in \mathcal{S}$, $\phi_s(\pi_0, \pi_1), \phi_s(\pi_1, \pi_2), \phi_s(\pi_2, \pi_0)$ are all less than 0.*

Proposition 1 shows that making assumptions on the preference distribution on trajectories cannot lead to a unique solution for PbRL. [6] Instead, since our target is an optimal policy, we make assumptions on the preferences between *trajectories*:

**Assumption 1.** *There exists a universal constant $C_0 > 0$ such that for any policies $\pi_1, \pi_2$ and state $s \in \mathcal{S}$ with $v_{\pi_1}(s) - v_{\pi_2}(s) > 0$, we have $\phi_s(\pi_1, \pi_2) \geq C_0(v_{\pi_1}(s) - v_{\pi_2}(s))$.*

I.e., the preference probabilities depends on the value function difference. Recall that $\phi_s(\pi_1, \pi_2)$ is the probability that a *random* trajectory from $\pi_1$ beats that from $\pi_2$; we do not make any assumptions on the comparison result of any *individual* trajectories. Assumption 1 ensures that a unique Condorcet winner (which is also the reward-maximizing policy) exists. Note that Assumption 1 also holds under many comparison models for $\phi_s$, such as the BTL or Thurstone model. Following previous literatures in dueling bandits [34, 17], we also define the following properties on preferences:

**Definition 2.** *Define the following properties on preferences when the following holds for any $h$, $s \in \mathcal{S}_h$ and three policies $\pi_1, \pi_2, \pi_3$ such that $v_h^{\pi_1}(s) > v_h^{\pi_2}(s) > v_h^{\pi_3}(s)$:*
***Strong Stochastic Transitivity:*** *$\phi_{s_h}(\pi_1, \pi_3) \geq \max\{\phi_{s_h}(\pi_1, \pi_2), \phi_{s_h}(\pi_2, \pi_3)\}$;*
***Stochastic Triangle Inequality:*** *$\phi_{s_h}(\pi_1, \pi_3) \leq \phi_{s_h}(\pi_1, \pi_2) + \phi_{s_h}(\pi_2, \pi_3)$.*

These properties are not essential for our algorithms, but are required for some dueling bandit algorithms that we use as a subroutine. To see the relation between policy preferences and reward preferences, we show the next proposition on several special cases:

**Proposition 2.** *If either of the following is true, the preferences satisfy Assumption 1, SST and STI:*
*i) There exists a constant $C'$ such that for every pair of trajectories $\tau, \tau'$ we have $\phi(\tau, \tau') = C'(r(\tau) - r(\tau'))$.*
*ii) The transitions are deterministic, and $\phi(\tau, \tau') = c$ for $r(\tau) > r(\tau')$ and some $c \in (0, 1/2]$.*

The first condition in Proposition 2 is the same as the assumption in [25], so our Assumption 1 is a generalization of theirs. We also note that although we focus on preferences over trajectories, preference between policies, as in Assumption 1, is also used in practice [18].

## 3 PbRL with a Simulator

In this section, we assume access to a simulator (generator) that allows access to any state $s \in \mathcal{S}$, executes an action $a \in \mathcal{A}$ and obtains the next state $s' \sim p(\cdot|s, a)$. We first introduce our algorithm, to then follow with theoretical results. We also show a lower bound that our comparison complexity is almost optimal.

Although value-based RL with a simulator has been well-studied in the literature [3, 4], methods like value iteration cannot easily extend to PbRL, because the reward values are hidden. Instead, we base our method on dynamic programming and policy search [7] - by running a dueling bandit algorithm on each state, one can determine the corresponding optimal action. The resulting algorithm, Preference-based Policy Search (PPS), is presented in Algorithm 1. By inducting from $H - 1$ to 0, PPS solves a dueling bandit problem at every state $s_h \in \mathcal{S}_h$, with arm rewards specified by $a \circ \hat{\pi}_{h+1:H}$ for $a \in \mathcal{A}$, where $\hat{\pi}$ is the estimated best policy after step $h + 1$, and $\circ$ stands for concatenation of policies. By allocating an error of $O(\varepsilon/H)$ on every state, we obtain the following guarantee:

**Algorithm 1** PPS: Preference-based Policy Search

---

**Require:** Dueling bandit algorithm $\mathcal{M}$, dueling accuracy $\varepsilon_1$, sampling number $N_2$, success probability $\delta$
  1: Initialize $\hat{\pi} \in \Pi^H$ randomly
  2: **for** $h = H - 1, ..., 0$ **do**
  3:    **for** $s_h \in \mathcal{S}_h$ **do**
  4:       **for** $n = 1, 2, ..., N_2$ **do**
  5:          Start an instance of $\mathcal{M}(\mathcal{A}, \varepsilon_1, \delta/S)$
  6:          Receive query $(a, a')$ from $\mathcal{M}$
  7:          Rollout $a \circ \hat{\pi}_{h+1:H}$ from $s_h$, get trajectory $\tau$
  8:          Rollout $a' \circ \hat{\pi}_{h+1:H}$ from $s_h$, get trajectory $\tau'$
  9:          Compare $\tau, \tau'$ and return the result to $\mathcal{M}$
 10:       **end for**
 11:       **if** $\mathcal{M}$ has finished **then**
 12:          Update $\hat{\pi}_h$ to the optimal action according to $\mathcal{M}$
 13:       **end if**
 14:    **end for**
 15: **end for**
**Ensure:** Policy $\hat{\pi}$

---

**Theorem 3.** *Suppose the preference distribution satisfies Assumption 1. Let $\varepsilon_1 = \frac{C_0 \varepsilon}{H}, N_0 = \Psi(\varepsilon_1, \delta/S)\varepsilon_1^{-\alpha}$, where $C_0$ is defined in Assumption 1. Algorithm 1 returns an $\varepsilon$-optimal policy with probability $1 - \delta$ using $O\left(\frac{H^{\alpha+1}S\Psi(A,\varepsilon/H,\delta/S)}{\varepsilon^\alpha}\right)$ simulator steps and $O\left(\frac{H^\alpha S\Psi(A,\varepsilon/H,\delta/S)}{\varepsilon^\alpha}\right)$ comparisons.*

We can plug in existing algorithms to instantiate Theorem 3. For example, under SST, OPT-Maximize [16] achieves the minimax optimal rate for dueling bandits. In particular, it has a comparison complexity of $O\left(\frac{K\log(1/\delta)}{\varepsilon^2}\right)$ for selecting an $\varepsilon$-optimal arm with probability $1 - \delta$ among $K$ arms. Plugging in this rate we obtain the following corollary:

**Corollary 4.** *Suppose the preference distribution satisfies Assumption 1 and SST. Using OPT-Maximize as $\mathcal{M}$, Algorithm 1 returns an $\varepsilon$-optimal policy with probability $1 - \delta$ using $O\left(\frac{H^3 SA}{\varepsilon^2}\log(S/\delta)\right)$ simulator steps, and $O\left(\frac{H^2 SA}{\varepsilon^2}\log(S/\delta)\right)$ comparisons.*

The proof of Theorem 3 follows from using the performance difference lemma, combined with properties of $\mathcal{M}$. Our result is similar to existing results for traditional RL with a simulator: For example, in the case of infinite-horizon MDPs with a decaying factor of $\gamma$, [5] shows a minimax rate of $O\left(\frac{SA}{\varepsilon^2(1-\gamma)^3}\right)$ on step complexity. This is the same rate as in Corollary 4 by taking $H = \frac{1}{1-\gamma}$ effective steps.

## 4   Combining Exploration and Policy Search for General PbRL

In this Section, we present our main result for PbRL without a simulator. RL without a simulator is a challenging problem even in the traditional value-based RL setting. In this case, we will have to explore the state space efficiently to find the optimal policy. Existing works in value-based RL typically derive an upper bound on the Q-value function and apply the Bellman equation to improve the policy iteratively [6, 22, 35]. However for PbRL, since the reward values are hidden, we cannot apply traditional value iteration and Q-learning methods. To go around this problem, we use a synthetic reward function to guide the exploration. We present our main algorithm in Section 4.1, along with the theoretical analysis. We discuss relations to prior work in Section 4.2.

### 4.1   Preferece-based Exploration and Policy Search (PEPS)

We call our algorithm Preference-based Exploration and Policy Search (PEPS), and present it in Algorithm 2. As the name suggests, the algorithm combines exploration and policy search. For every

---

**Algorithm 2** PEPS: Preferece-based Exploration and Policy Search

---

**Require:** Dueling Bandit algorithm $\mathcal{M}$, quantity $N_0$, success probability $\delta$
1: Initialize $\hat{\pi} \in \Pi^H$ randomly
2: **for** $h = H, H-1, ..., 1$ **do**
3:      **for** $s_h \in \mathcal{S}_h$ **do**
4:          Let $r_{s_h}(s, a) = 1_{s=s_h}$ for all $s \in S, a \in A$
5:          Start an instance of EULER$(r_{s_h}, N_0, \delta/(4S))$
6:          Start an instance of $\mathcal{M}(\mathcal{A}, \delta/(4S))$
7:          Get next query $(a, a')$ from $\mathcal{M}$
8:          $U \leftarrow \emptyset$
9:          **for** $n \in [N_0]$ **do**
10:              Obtain a policy $\hat{\pi}_n$ from EULER, and execute $\hat{\pi}_n$ until step $h$
11:              Return the trajectory and reward to EULER
12:              **if** current state is $s_h$ **then**
13:                  Let $\bar{\pi} = a \circ \hat{\pi}_{h+1:H}$ if $|U| = 0$, otherwise $a' \circ \hat{\pi}_{h+1:H}$
14:                  Execute $\bar{\pi}$ till step $H$, obtain trajectory $\tau$
15:                  $U \leftarrow U \cup \tau$
16:              **end if**
17:              **if** $|U| = 2$ **then**
18:                  Compare the two trajectories in $U$ and return to $\mathcal{M}$
19:                  $(a, a') \leftarrow$ next action from $\mathcal{M}$
20:              **end if**
21:              If $\mathcal{M}$ has finished, break
22:          **end for**
23:          **if** $\mathcal{M}$ has finished **then**
24:              Update $\hat{\pi}_h(s_h)$ to the optimal action according to $\mathcal{M}$
25:          **end if**
26:      **end for**
27: **end for**
**Ensure:** Policy $\hat{\pi}$

---

step $h \in [H]$ and $s_h \in \mathcal{S}_h$, we set up an artificial reward function $r_{s_h}(s, a) = 1_{s=s_h}$, i.e., the agent gets a reward of 1 once it gets to $s_h$, and 0 everywhere else (Step 4). This function is also used in recent reward-free learning literatures [14, 24, 23]. But rather than using the reward function to obtain a policy cover (which is costly in both time and space), we use it to help the dueling bandit algorithm. We can use arbitrary tabular RL algorithm to optimize $r_{s_h}$; here we use EULER [35] with a budget of $N_0$ and success probability $\delta/4S$. The reason that we chose EULER is mainly for its beneficial regret guarantees; but we can also use other algorithms in practice. We also start an instance of $\mathcal{M}$, the dueling bandit algorithm, without setting the target accuracy; one way to achieve this is to use a pre-defined accuracy for $\mathcal{M}$, or use algorithms like Beat-the-Mean (see Theorem 5 and 7 respectively).

Once we get to $s_h$ (Step 12), we execute the queried action $a$ or $a'$ from $\mathcal{M}$, followed by the current best policy $\hat{\pi}$, as in PPS. If we have collected two trajectories, we compare them and feed it back to $\mathcal{M}$. Upon finishing $N_0$ steps of exploration, we update the current best policy $\hat{\pi}$ according to $\mathcal{M}$. We return $\hat{\pi}$ when we have explored every state $s \in \mathcal{S}$.

Throughout this section, we use $\iota = \log\left(\frac{SAH}{\varepsilon\delta}\right)$ to denote log factors. We present two versions of guarantees with different sample complexity. The first version has a lower comparison complexity, while the second version has a lower total step complexity. The different guarantee comes from setting a slightly different target for $\mathcal{M}$ when it finishes in Step 21. In the following first version, we set $\mathcal{M}$ to finish when it finds a $O(\varepsilon/H)$-optimal policy.

**Theorem 5.** *Suppose the preference distribution satisfies Assumption 1. There exists a constant $c_0$ such that the following holds: Let $N_0 = c_0 \left( \frac{H^\alpha S\Psi(A, \varepsilon/H, \delta/4S)}{\varepsilon^{\alpha+1}} + \frac{S^3 AH^2 \iota^3}{\varepsilon} \right)$, recall $\iota = \log\left(\frac{SAH}{\varepsilon\delta}\right)$. By setting the target accuracy as $\frac{C_0 \varepsilon}{2H}$ for $\mathcal{M}$, PEPS obtains an $\varepsilon$-optimal policy with probability*

$1 - \delta$ using step complexity

$$O(HSN_0) = O\left(\frac{H^{\alpha+1}S^2\Psi(A, \varepsilon/H, \delta/4S)}{\varepsilon^{\alpha+1}} + \frac{S^4AH^3\iota^3}{\varepsilon}\right)$$

and comparison complexity $O\left(\frac{H^\alpha S\Psi(A, \varepsilon/H, \delta/4S)}{\varepsilon^\alpha}\right)$.

Since we set the target accuracy before the algorithm starts, any PAC dueling bandit algorithm can be used to instantiate Theorem 5. So similar to Theorem 3, we can plug in OPT-Maximize [16] to obtain the following corollary:

**Corollary 6.** *Suppose the preference distribution satisfies Assumption 1 and SST. There exists a constant $c_0$ such that the following holds: Let $N_0 = c_0\left(\frac{SH^2A\log(S/\delta)}{\varepsilon^3} + \frac{S^3AH^2\iota^3}{\varepsilon}\right)$. Using OPT-Maximize with accuracy $\varepsilon/H$ as $\mathcal{M}$, PEPS obtains an $\varepsilon$-optimal policy with probability $1 - \delta$ using step complexity*

$$O(HSN_0) = O\left(\frac{H^3S^2A\log(S/\delta)}{\varepsilon^3} + \frac{S^4AH^3\iota^3}{\varepsilon}\right)$$

and comparison complexity $O\left(\frac{H^2SA\log(S/\delta)}{\varepsilon^2}\right)$.

In the second version, we simply do not set *any* performance target for $\mathcal{M}$; it will explore until we finish all the $N_0$ episodes. Therefore, we never break in Step 21 and $\mathcal{M}$ is always finished in Step 23. Since different states will be reached for a different number of times, we cannot pre-set a target accuracy for $\mathcal{M}$. Effectively, we explore every state $s$ to the accuracy of $\varepsilon_s = O\left(\frac{\varepsilon}{(\mu(s_h)SH^{\alpha-1})^{1/\alpha}}\right)$ (see proof in appendix for details), where $\mu(s)$ is the maximum probability to reach $s$ using any policy. Although this version leads to a slightly higher comparison complexity, it leads to a better step complexity since all exploration steps are used efficiently. We have the following result:

**Theorem 7.** *Suppose the preference distribution satisfies Assumption 1. There exists a constant $c_0$ such that the following holds: Let $N_0 = c_0\left(\frac{H^{\alpha-1}S\Psi(A, \frac{\varepsilon}{HS}, \delta/4S)}{\varepsilon^\alpha} + \frac{S^3AH^2\iota^3}{\varepsilon}\right)$. PEPS obtains an $\varepsilon$-optimal policy with probability $1 - \delta$ using step complexity*

$$O(HSN_0) = \left(\frac{H^\alpha S^2\Psi(A, \frac{\varepsilon}{HS}, \delta/4S)}{\varepsilon^\alpha} + \frac{S^4AH^3\iota^3}{\varepsilon}\right)$$

and comparison complexity $O\left(\frac{H^{\alpha-1}S^2\Psi(A, \frac{\varepsilon}{HS}, \delta/4S)}{\varepsilon^\alpha}\right)$.

We need to instantiate Theorem 7 with an $\mathcal{M}$ that does not need a pre-set accuracy. Under SST and STI, we can use Beat-the-Mean [34], which returns an $\varepsilon$-optimal arm among $K$ arms with probability $1 - \delta$, if it runs for $\Omega\left(\left(\frac{K}{\varepsilon^2}\log\left(\frac{K}{\varepsilon\delta}\right)\right)\right)$ steps. We therefore obtain the following corollary:

**Corollary 8.** *Suppose the preference distribution satisfies Assumption 1, SST and STI. There exists a constant $c_0$ such that the following holds: Let $N_0 = c_0\left(\frac{HSA\iota}{\varepsilon^2} + \frac{S^3AH^2\iota^3}{\varepsilon}\right)$. Using Beat-the-Mean as $\mathcal{M}$, PEPS obtains an $\varepsilon$-optimal policy with probability $1 - \delta$ using step complexity*

$$O(HSN_0) = O\left(\frac{H^2S^2A\iota}{\varepsilon^2} + \frac{S^4AH^3\iota^3}{\varepsilon}\right)$$

and comparison complexity $O\left(\frac{HS^2A\iota}{\varepsilon^2}\right)$.

## 4.2   Discussion

**Comparing Corollary 6 and 8.** Considering only the leading terms, the step complexity in Corollary 8 is better by a factor[7] of $\tilde{O}(H/\varepsilon)$ but the comparison complexity is worse by a factor of $\tilde{O}(S/H)$ (recall that we assume $S > H$). Therefore the two theorems depict a tradeoff between the step complexity and comparison complexity.

**Comparing with lower bounds and value-based RL.** Corollary 6 has the same comparison complexity as Corollary 4. The lower bound for comparison complexity is $\tilde{O}(\frac{SA}{\varepsilon^2})$, a $\tilde{O}(H^2)$ factor from our result. Our comparison complexity is also the same as the current best upper bound in value-based RL (in terms of number of episodes), which shows that our result is close to optimal.

Compared with the $\tilde{O}(\frac{HSA}{\varepsilon^2})$ lower bound on step complexity, Corollary 8 has an additional factor of $\tilde{O}(HS)$. The current best upper bound in value-based RL is $\tilde{O}(\frac{H^3SA}{\varepsilon^2})$, which is $O(H/S)$ times our result (recall that we assume $H < S$).

**Comparing with previous RL method using policy covers.** Some literature on reward-free learning and policy covers [14, 24, 23] use a similar form of synthetic reward function as ours. [14, 24] considers computing a policy cover by exploring the state space using a similar synthetic function as PEPS. However, our results are generally not comparable since they assume that $\mu(s) \geq \eta$ for some $\eta > 0$, and their result depends on $\eta$. Our result do not need to depend on this $\eta$. Closest to our method is the recent work of [23], which considers reward-free learning with unknown rewards during exploration. However, their method cannot be applied to PbRL since we do not have the reward values even in the planning phase. We note that the $O(S^2)$ dependence on the step complexity is also common in these prior works. Nevertheless, our rates are better in $H$ dependence because we do not need to compute the policy cover explicitly.

**Fixed budget setting.** While we present PPS and PEPS under the fixed confidence setting (with a given $\varepsilon$), one can easily adapt it to the fixed budget setting (with a given $N$, number of episodes) by dividing $N$ evenly among the states. We present a fixed budget version in the appendix.

**Two-phase extension of PEPS.** A drawback of Theorem 7 is that it requires $\mathcal{M}$ to work without a pre-set target accuracy, limiting the algorithm that we can use to instantiate $\mathcal{M}$. In the appendix, we present a two-phase version of PEPS that allows for arbitrary PAC algorithm $\mathcal{M}$, with slightly improved log factors on the guarantee.

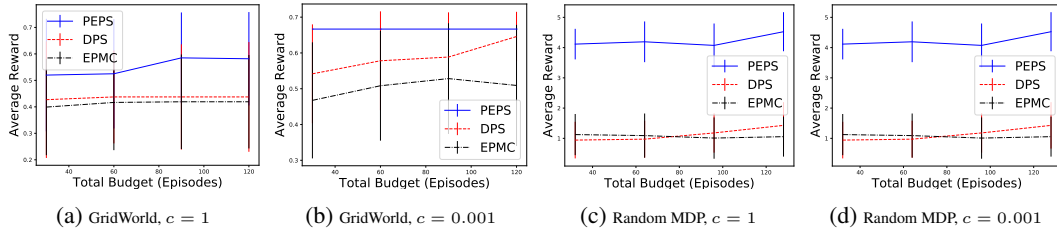

(a) GridWorld, $c = 1$     (b) GridWorld, $c = 0.001$     (c) Random MDP, $c = 1$     (d) Random MDP, $c = 0.001$

Figure 1: Experiment Results comparing PEPS to baselines (DPS & EPMC).

## 5   Experiments

We performed experiments in synthetic environments to compare PEPS with previous baselines. We consider two environments:

*Grid World*: We implemented a simple Grid World on a $4 \times 4$ grid. The agent goes from the upper left corner to the lower right corner and can choose to go right or go down at each step. We randomly put a reward of $1/3$ on three blocks in the grid, and the maximal total reward is $2/3$.

*Random MDP*: We followed the method in [25] but adapted it to our setting. We consider an MDP with 20 states and 5 steps, with 4 states in each step. The transitions are sampled from a Dirichlet prior (with parameters all set to 0.1) and the rewards are sampled from an exponential prior with scale parameter $\lambda = 5$. The rewards are then shifted and normalized so that the minimum reward is 0 and the mean reward is 1.

**Compared methods.** We compared to three baselines: i) DPS [25]: We used the version with Gaussian process regression since this version gets the best result in their experiments; ii) EPMC [30], which uses preferences to simulate a Q-function. We followed the default parameter settings for both DPS and EPMC. Details of the algorithms and hyperparameter settings are included in the appendix.

**Experiment Setup.** Our baselines are not directly comparable to PEPS since their goal is to minimize the regret, instead of getting an $\varepsilon$-optimal policy. However, we can easily adapt all the algorithms (including PEPS) to the fixed budget setting for optimal policy recovery. For the baselines, we ran

them until $N$ episodes and then evaluated the current best policy. For PEPS, we used the fixed budget version described in detail in the appendix. For both environments, we varied the budget $N \in [2S', 8S']$, where $S'$ is the number of non-terminating states. The comparisons are generated following the Bradley-Terry-Luce model [9]: $\phi(\tau_1, \tau_2) = \frac{1}{1+\exp(-(r(\tau_1)-r(\tau_2))/c)}$, with $c$ being either 0.001 or 1. In the first setting of $c$, the preferences are very close to deterministic while comparison between equal rewards is uniformly random; in the latter setting, the preferences are close to linear in the reward difference. We repeated each experiment for 32 times and computed the mean and standard deviation.

**Results.** The results are summarized in Figure 1. Overall, PEPS outperforms both baselines in both environments. In Grid World, while all three methods get a relatively high variance when $c = 1$, for $c = 0.001$ PEPS almost always get the exact optimal policy. Also for random MDP, PEPS outperforms both baselines by a large margin (larger than two standard deviations). We note that EPMC learns very slowly and almost does not improve as the budget increases, and this is consistent with the observation in [25]. One potential reason that makes PEPS outperform the baselines is because of the exploration method: Both DPS and EPMC need to estimate the Q function well in order to perform efficient exploration. This estimation can take time exponential in $H$, and it is not even computationally feasible to test until the Q function converges. As a result, both DPS and EPMC explore almost randomly and cannot recover the optimal policy. On the other hand, our method uses a dueling bandit algorithm to force exploration, so it guarantees that at least states with high reach probability have their optimal action.

## 6 Conclusion

We analyze the theoretical foundations of the PbRL framework in detail and present the first finite-time analysis for general PbRL problems. Based on reasonable assumptions on the preferences, the proposed PEPS method recovers an $\varepsilon$-optimal policy with finite-time guarantees. Experiments show the potential efficacy of our method in practice, and that it can work well in simulated environments. Future work includes i) testing PEPS in other RL environments and applications, ii) developing algorithms for PbRL with finite-time regret guarantees, iii) algorithms for the case when Assumption 1 holds with some error, and iv) PbRL in non-tabular settings.

## Broader Impact Discussion

Reinforcement learning is increasingly being used in a myriad of decision-making applications ranging from robotics and drone control, to computer games, to tutoring systems, to clinical trials in medicine, to social decision making. Many of these applications have goals that are hard to capture mathematically using a single reward definition, and the common practice of reward hacking [8, 1] (trying to achieve best performance under the specified metric) often leads to undesired behaviors with respect to the original goal of the application. Preference based RL provides an appealing alternative where the user can specify their preferences over alternative trajectories without hard-coding a single reward metric and this way avoid unexpected behavior caused by misspecified reward metrics. This paper substantially expands theoretical understanding of preference based RL by providing the first finite time guarantees, and it could help broaden applicability of this learning modality across multiple areas of its potential beneficial use, in particular where it is currently considered too expensive to set up and deploy.

## Footnotes

*Work done while at Carnegie Mellon University.

[2]For simplicity, here we assume the reward function to be deterministic.

[3]We consider deterministic policies for simplicity, but our results carry to random policies easily.

[4]We follow the standard notation to denote $[H] = \{0, 1, ..., H-1\}$.

[5]Note that some prior works assumes that $r(s, a) \in [0, 1/H]$ [6, 35], which is different from our setting.

[6]One might consider other winner notions when a Condorcet winner policy does not exist. e.g., the von Neumann winner [15]. However, finding such winners usually involves handling an exponential number of policies and is out of the scope of the current paper.

[7] We use $\tilde{O}$ to ignore log factors.

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
