[Supplementary Material]

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

# A  Another Version to Accommodate Arbitrary PAC Dueling Algorithm

A drawback of PEPS is that the version in Theorem 7 requires a dueling algorithm $\mathcal{M}$ that does not need a target accuracy, restricting the possible types of algorithms we can use. In this section, we present a two-phase version of our algorithm to allow arbitrary $\mathcal{M}$.

The two-phase version is presented in Algorithm 3 as PEPS2. PEPS2 separates the process of obtaining a policy cover and using dueling bandits for policy search; in the first phase (Step 1-9) uses the synthetic reward function to obtain a policy cover with EULER. We then estimate $\mu(s)$ for every state $s \in \mathcal{S}$ by executing the policy that we obtain. Using the estimate $\hat{\mu}_s$, we follow the process in PEPS with the target accuracy specified in Step 12.

---

**Algorithm 3** PEPS2: Preferece-based Exploration and Policy Search with 2 phases

---

**Require:** Target Accuracy $\varepsilon$, Dueling Bandit algorithm $\mathcal{M}$, quantities $N_0, N_1, N_2$, success probability $\delta$

1: **for** $h \in [H]$ **do**
2: &emsp; $\tilde{S}_h = \emptyset$
3: &emsp; **for** $s_h \in \mathcal{S}_h$ **do**
4: &emsp;&emsp; Let $r_{s_h}(s,a) = 1_{s=s_h}$ for all $s \in S, a \in A$
5: &emsp;&emsp; Obtain a policy $\hat{\pi}_{s_h}$ by using EULER$(r_{s_h}, N_0, \delta/(4S))$ to optimize $r_{s_h}(s,a)$
6: &emsp;&emsp; Rollout $\hat{\pi}_{s_h}$ for $N_1$ times and record reward the total reward $\hat{R}_{s_h}$ under $r_{s_h}$
7: &emsp;&emsp; Let $\hat{\mu}_{s_h} \leftarrow \min\{1, 2\hat{R}_{s_h}/N_1 + 2\sqrt{\frac{\log(4S/\delta)}{N}}\}$
8: &emsp; **end for**
9: **end for**
10: Initialize $\hat{\pi} \in \Pi^H$ randomly
11: **for** $h = H, H-1, ..., 1$ **do**
12: &emsp; Let $\varepsilon_{s_h} \leftarrow \frac{\varepsilon}{4(\hat{\mu}(s_h)SH^{\alpha-1})^{1/\alpha}}$
13: &emsp; **for** $s_h \in \mathcal{S}_h$ **do**
14: &emsp;&emsp; Start an instance of $\mathcal{M}(\mathcal{A}, \varepsilon_{s_h}, \delta/(4S))$
15: &emsp;&emsp; Get next query $(a, a')$ from $\mathcal{M}$
16: &emsp;&emsp; $U \leftarrow \emptyset$
17: &emsp;&emsp; **for** $n \in [N_0]$ **do**
18: &emsp;&emsp;&emsp; Execute $\hat{\pi}_{s_h}$ until step $h$
19: &emsp;&emsp;&emsp; **if** current state is $s_h$ **then**
20: &emsp;&emsp;&emsp;&emsp; Let $\bar{\pi} = a \circ \hat{\pi}_{h+1:H}$ if $|U| = 0$, otherwise $a' \circ \hat{\pi}_{h+1:H}$
21: &emsp;&emsp;&emsp;&emsp; Execute $\bar{\pi}$ till step $H$, obtain trajectory $\tau$
22: &emsp;&emsp;&emsp;&emsp; $U \leftarrow U \cup \tau$
23: &emsp;&emsp;&emsp; **end if**
24: &emsp;&emsp;&emsp; **if** $|U| = 2$ **then**
25: &emsp;&emsp;&emsp;&emsp; Compare the two trajectories in $U$ and return to $\mathcal{M}$
26: &emsp;&emsp;&emsp; **end if**
27: &emsp;&emsp;&emsp; If $\mathcal{M}$ has finished, break
28: &emsp;&emsp; **end for**
29: &emsp;&emsp; **if** $\mathcal{M}$ has finished **then**
30: &emsp;&emsp;&emsp; Update $\hat{\pi}_h(s_h)$ to the optimal action according to $\mathcal{M}$
31: &emsp;&emsp; **end if**
32: &emsp; **end for**
33: **end for**
**Ensure:** Policy $\hat{\pi}$

---

For every state $s \in \mathcal{S}$, let $\mu(s) = \max_{\pi \in \Pi} p_\pi(s)$ be the maximum probability to reach $s$.

**Theorem 9.** *There exists a constant $c_0$ such that the following holds. Let $N_0 = c_0 \frac{S^3 A H^2 \iota^3}{\varepsilon}, N_1 = \frac{c_0 S^2 \log(4S/\delta)}{\varepsilon^2}, N_2 = \frac{H^{\alpha-1}S\Psi(A, \frac{\varepsilon}{HS}, \delta/4S)}{\varepsilon^\alpha}$. With probability $1 - \delta$, Algorithm 3 returns an $\varepsilon$-optimal policy using $HS(N_0 + N_1 + N_2)$ steps and $SN_2$ comparisons.*

We can plug in the guarantee of OPT-Maximize instantiate Theorem 9. This result is better in terms of the log terms than the result in Corollary 8.

**Corollary 10.** *There exists constants $c_0$ such that the following holds: Let $N_0 = c_0 \left( \frac{HSA \log(\delta/S)}{\varepsilon^2} + \frac{S^3 AH^2 \iota^3}{\varepsilon} \right)$, where $\iota = \log(\frac{SAH}{\varepsilon\delta})$. Using OPT-Maximize as $\mathcal{M}$, PEPS2 obtains an $\varepsilon$-optimal policy with probability $1 - \delta$ using step complexity*

$$O(HSN_0) = O\left( \frac{H^2 S^2 A \log(\delta/S)}{\varepsilon^2} + \frac{S^4 AH^3 \iota^3}{\varepsilon} \right)$$

*and comparison complexity $O\left( \frac{HS^2 A \log(S/\delta)}{\varepsilon^2} \right)$.*

## B   Adapting PEPS to the Fixed Budget setting

We adapt PEPS to the fixed budget setting, described in detail in Algorithm 4. To do this, we set $N_0 = N/S$, where $S$ is the number of non-terminating states. Before the algorithm start, we start an instance of $\mathcal{M}$ for every state $s \in \mathcal{S}$; and instead of only compare when reaching the target state, we simply explore according to $\mathcal{M}$ regardless of the current state at step $h$.

In our experiments We realize PEPS with Q-learning instead of EULER because of its computational efficiency; we use the formulation of [22] with a Hoeffding upper bound on the Q function. For $\mathcal{M}$, we use Beat-the-Mean because it can use any budget.

---
**Algorithm 4** PEPS with Fixed Budget
---
**Require:** Budget $N$, dueling bandit algorithm $\mathcal{M}$, success probability $\delta$
 1: Initialize $\hat{\pi} \in \Pi^H$ randomly
 2: Start an instance of $\mathcal{M}(\mathcal{A}, \delta/(4S))$ at every state $s \in \mathcal{S}$ (denote it by $\mathcal{M}_s$), and get first action $(a_s, a_s')$
 3: **for** $h = H, H-1, ..., 1$ **do**
 4:      **for** $s_h \in \mathcal{S}_h$ **do**
 5:          Let $r_{s_h}(s, a) = 1_{s=s_h}$ for all $s \in S, a \in A$
 6:          Start an instance of $\text{EULER}(r_{s_h}, N_0, \delta/(4S))$
 7:          $U \leftarrow \emptyset$
 8:          **for** $n \in [N/S]$ **do**
 9:              Obtain a policy $\hat{\pi}_n$ from EULER, and execute $\hat{\pi}_n$ until step $h$
10:              Return the trajectory and reward to EULER
11:              $\tilde{s} \leftarrow$ current state
12:              Let $\bar{\pi} = a_{\tilde{s}} \circ \hat{\pi}_{h+1:H}$ if $|U| = 0$, otherwise $a_{\tilde{s}}' \circ \hat{\pi}_{h+1:H}$
13:              Execute $\bar{\pi}$ till step $H$, obtain trajectory $\tau$
14:              $U \leftarrow U \cup \tau$
15:              **if** $|U| = 2$ **then**
16:                  Compare the two trajectories in $U$ and return to $\mathcal{M}_{\tilde{s}}$
17:                  Get next action $(a_{\tilde{s}}, a_{\tilde{s}}')$ from $\mathcal{M}_{\tilde{s}}$
18:                  Update $\hat{\pi}_h(\tilde{s})$ to the optimal action according to $\mathcal{M}_{\tilde{s}}$
19:              **end if**
20:          **end for**
21:      **end for**
22: **end for**
**Ensure:** Policy $\hat{\pi}$
---

## C   Additional Experiment Details

### C.1   Hyperparameters for Experiments

For PEPS, we search the hyperparameters for Q learning and Beat-the-Mean. This includes the learning rate of Q-learning (in range $\{0.1, 0.3, 1.0\}$), the ucb bound ratio for Q-learning (in range $\{0.01, 0.1, 1.0\}$), and the ucb bound ratio for Beat-the-Mean (in range $\{0.2, 0.5, 1.0\}$). We also allow the algorithm to random explore with probability in $\{0.05, 0.1, 0.2, 0.5\}$ during Q-learning. For DPS, we follow the default settings to use kernel variance 0.1 and kernel noise 0.001 for the Gaussian process regression. For EPMC, we use $\alpha = 0.2$ and $\eta = 0.8$.

(a) Linear, GridWorld  (b) Linear, Random MDP  (c) Deterministic, GridWorld  (d) Deterministic, Random MDP

Figure 2: Experiment results under linear preferences (a,b) and deterministic preferences (c,d).

(a) Varying $c$ in BTL model

(b) Varying $H$

Figure 3: Performance dependence on BTL model parameter $c$, and time horizon $H$.

## C.2 Additional Experiment results

**Results under other preference models.** We test the performance of PEPS and other baselines under the linear preference model and deterministic preferences in Figure 2. For linear preferences, we generate the preferences through a linear link function $\phi(\tau_1, \tau_2) = \alpha(r(\tau_1) - r(\tau_2))$, and we use $\alpha = 0.01$ in our experiment. This results in a very noisy preference model. PEPS performs similarly to the baselines in GridWorld, potentially because of the noisy preferences; but it still outperforms the baselines in random MDP. Under deterministic preferences, PEPS outperforms both baselines in both environments. **Dependence on parameters.** We test the performance versus various parameters in Figure 3. In Figure 3a we vary the BTL model parameter $c$. Our PEPS consistently outperforms the baselines. In Figure 3b we test the regret versus the time horizon $H$. It shows a close to linear relation, which fits our rate in Corollary 8 (the $O(H^2/\varepsilon^2)$ term translates to a linear dependence of $\varepsilon$ on $H$).

# D  Proofs

## D.1  Proof of Proposition 1

*Proof.* Let $S = 6$, $\mathcal{S} = \{s_0, ..., s_5\}$, and $A = 3$, $\mathcal{A} = \{a_0, a_1, a_2\}$, and let $H = 2$. We start at $s_0$. Let $r(s_0, a) = 0$ for all $a \in \mathcal{A}$. Executing $a_0$ in $s_0$ goes to $s_1$ w.p. 0.2, and to $s_2$ w.p. 0.8. Executing $a_1$ in $s_0$ goes to $s_3$ with probability 1. Executing $a_2$ in $s_0$ goes to $s_4$ w.p. 0.6 and to $s_5$ w.p. 0.4. For every action $a$, let $r(s_1, a) = 1, r(s_2, a) = 0.01, r(s_3, a) = 0.02, r(s_4, a) = 0.5, r(s_5, a) = 0$. See Figure 4 for a graphical explanation.

Let $\pi_i(s_0) = a_i$ for $i \in \{0, 1, 2\}$ (actions in other states do not matter). It is easy to verify that $\phi_{s_0}(\pi_0, \pi_1) = -0.3, \phi_{s_0}(\pi_1, \pi_2) = -0.1$, and $\phi_{s_0}(\pi_2, \pi_0) = -0.02$. □

Figure 4: Example for proof of Proposition 1.

## D.2 Proof of Proposition 2

*Proof.* For i), by the linearity of expectation we have for two random trajectories $\tau, \tau'$, we have $E[\phi(\tau, \tau')] = E[C(r(\tau) - r(\tau'))]$. Therefore for two policies $\pi_1, \pi_2$

$$\phi_s(\pi_1, \pi_2) = \Pr[\tau(\pi_1, s) \succ \tau(\pi_2, s)] - 1/2$$
$$= E[\phi(\tau(\pi_1, s), \tau(\pi_2, s))]$$
$$= E[C(r(\tau(\pi_1, s)) - r(\tau(\pi_2, s)))] = C(v_s(\pi_1) - v_s(\pi_2)).$$

For ii), when transitions are deterministic each policy corresponds to only one trajectory. Let $\tau_1$ and $\tau_2$ be the two trajectories corresponding to $\pi_1$ and $\pi_2$ starting at $s$. Then we have $\phi_s(\pi_1, \pi_2) = \phi(\tau_1, \tau_2)$. Then Assumption 1 is satisfied with $C_0 = c$. □

## D.3 Proof of Theorem 3

*Proof.* We only need to show that the output policy $\hat{\pi}$ is $\varepsilon$-optimal. Algorithm 1 loops over $h = 1, 2, ..., H$. At every step $h$ for every state $s \in \mathcal{S}_h$, let $\tilde{\pi}^*(s) = \arg\max_a V(s; a \circ \hat{\pi}_{h+1:H})$. Let $P_h^*$ denote the state distribution of $\pi^*$ at step $h$. With probability $1 - \delta$, all instances of $\mathcal{M}$ has finished. Under this event, from the property of $\mathcal{M}$ and the setup of $N_0$ we have for every state $s \in \mathcal{S}$, $\Pr[\phi_s(\hat{\pi}, \tilde{\pi}^*(s))] \geq 1/2 - \varepsilon_1$. Therefore from Assumption 1 we have

$$|v_{\tilde{\pi}^*(s)}(s) - v_{\hat{\pi}}(s)| \leq \frac{1}{C_0}\phi_s(\tilde{\pi}^*(s), \hat{\pi}) \leq \frac{\varepsilon}{H}.$$

Therefore using the performance difference lemma we have

$$V^{\pi^*}(s_0) - V^{\hat{\pi}}(s_0) = \sum_{h=1}^{H} E_{s_h \sim P_h^*}[v(s_h; \pi_h^* \circ \hat{\pi}_{h+1:H}) - v(s_h; \hat{\pi}_{h:H})]$$

$$\leq \sum_{h=1}^{H} E_{s_h \sim P_h^*}[v(s_h; \tilde{\pi}_h^* \circ \hat{\pi}_{h+1:H}) - v(s_h; \hat{\pi}_{h:H})]$$

$$\leq \sum_{h=1}^{H} \frac{\varepsilon}{H} = \varepsilon.$$

□

## D.4 Proof of Theorem 5

*Proof.* Let $\tilde{\mathcal{S}}_h = \{s \in \mathcal{S}_h | \mu(s) \geq \varepsilon/2S\}$ be the set of "good" states that are reachable with probability at least $\varepsilon/(2S)$. Also let $\tilde{\mathcal{S}} = \bigcup_h \tilde{\mathcal{S}}_h$ be the set of all good states.

Now we show that the output policy $\hat{\pi}$ is $\varepsilon$-optimal. We first present the performance of EULER [35]:

**Theorem 11** (Theorem 1, [35]). *Let $Q^*$ be the value of the optimal policy. For any reward function $r$, the regret of EULER is at most*

$$R \leq O(\sqrt{Q^* SATL} + \sqrt{S}SAH^2L^3(\sqrt{S} + \sqrt{H}))$$

*with probability $1 - \delta$, with $L = \log(HSAT/\delta)$.*

For any state $s$, let $N_s$ be the number of times that we reach $s$ in Step 12. Using Theorem 11 with $T = HN_0$, we have for some constant $C$,

$$\mu(s)N_0 - N_s \le C(\sqrt{\mu(s)SAHN_0L} + \sqrt{S}SAH^2L^3(\sqrt{S} + \sqrt{H})) \qquad (1)$$

with probability $1 - \delta/4S$. By a union bound, suppose (1) holds for every state $s \in \mathcal{S}$, which happens with probability $1 - \delta/4$. Now consider any $s \in \tilde{\mathcal{S}}$. With $N_0 = \Omega(\frac{S^3AH^2\iota^3}{\varepsilon})$, we have $N_s \ge 1/2\mu(s)N_0$. Now also using $N_0 = \Omega\left(\frac{SH^\alpha\Psi(A,\varepsilon/H,\delta/4S)}{\varepsilon^{\alpha+1}}\right)$, we have

$$N_s \ge \frac{1}{2}\mu(s)N_0 \ge \Omega\left(\frac{H^\alpha\Psi(A,\varepsilon/H,\delta/4S)}{\varepsilon^\alpha}\right).$$

By setting the constant in $N_0$ properly, from the definition of $\Psi$ we know that with probability $1 - \delta/4S$, $\mathcal{M}$ returns a $\frac{c_K\varepsilon}{H}$ optimal arm; here $c_K$ is a constant to be specified later.

Algorithm 2 loops over $h = 1, 2, ..., H$. At every step $h$ for every state $s \in \mathcal{S}_h$, let $\tilde{a}_s^* = \arg\max_a V(s; a \circ \hat{\pi}_{h+1:H})$. From the guarantees of OPT-Maximize we have

$$v(s_h; \tilde{a}_s^* \circ \hat{\pi}_{h+1:H}) - v(s_h; \hat{\pi}_{h:H}) \le \frac{1}{C_0}\phi_s(\tilde{a}_s^*, \hat{\pi}_{h+1:H}) \le \frac{\varepsilon}{2H}.$$

The first inequality comes from Assumption 1; we set $c_K$ small enough to satisfy the latter inequality.

Let $P_h^*$ denote the state distribution of $\pi^*$ at step $h$. We have

$$V^{\pi^*}(s_0) - V^{\hat{\pi}}(s_0) = \sum_{h=1}^H E_{s_h \sim P_h^*}[v(s_h; \pi_h^* \circ \hat{\pi}_{h+1:H}) - v(s_h; \hat{\pi}_{h:H})]$$

$$\le \sum_{h=1}^H E_{s_h \sim P_h^*}[v(s_h; \tilde{\pi}_h^* \circ \hat{\pi}_{h+1:H}) - v(s_h; \hat{\pi}_{h:H})]$$

$$\le \sum_{h=1}^H \Pr_{\pi^*}[s_h \notin \tilde{\mathcal{S}}] + \Pr[s_h \in \tilde{\mathcal{S}}]E_{s_h \sim P_h^*, s_h \in \tilde{\mathcal{S}}}[v(s_h; \tilde{\pi}_h^* \circ \hat{\pi}_{h+1:H}) - v(s_h; \hat{\pi}_{h:H})]$$

$$\le \varepsilon/(2S) \cdot S + \sum_{h=1}^H \varepsilon/(2H) \le \varepsilon.$$

Here the first equality is the performance difference lemma (Lemma 13, [24]). The first inequality comes from the definition of $\tilde{\pi}^*$; the third inequality comes from definition of $\tilde{\mathcal{S}}$, and the guarantee of $\mathcal{M}$. Therefore we show that the output policy is $\varepsilon$-optimal.

Now we compute the sample complexity. It is obvious that the step complexity is $HSN_0$ since we iterate through all $s \in \mathcal{S}$, and each episode contains $H$ steps. For comparison complexity, we only need to finish $S$ instances of $\mathcal{M}$; therefore the comparison complexity is $O\left(\frac{H^\alpha S\Psi(A,\varepsilon/H,\delta/4S)}{\varepsilon^\alpha}\right)$.

$\square$

### D.5 Proof of Theorem 7

*Proof.* The proof follows most parts of that of 5. For any $s \in \tilde{\mathcal{S}}$, we have $N_s \ge \frac{1}{2}\mu(s)N_0$. Guarantee of Beat-the-Mean is as follows:

For simplicity, let $p = 1/\alpha$ and $q = (\alpha - 1)/\alpha$. For $s_h \in \tilde{\mathcal{S}}_h$, let $\varepsilon_{s_h} = \frac{\varepsilon}{2\mu^p(s_h)S^pH^q}$. For $N_0 = \Omega\left(\frac{H^{\alpha-1}S\Psi(A,\frac{\varepsilon}{HS},\delta/4S)}{\varepsilon^\alpha}\right)$, we have

$$N_{s_h} \ge 1/2\mu(s_h)N_0 = \Omega\left(\frac{\mu(s_h)H^{\alpha-1}S\Psi(A,\frac{\varepsilon}{HS},\delta/4S)}{\varepsilon^\alpha}\right) \ge \Omega\left(\Psi(A,\varepsilon_{s_h},\delta/4S)\varepsilon_{s_h}^{-\alpha}\right).$$

Similar to proof of Theorem 5, we can set the constants properly to obtain that

$$v(s_h; \tilde{\pi}_h^* \circ \hat{\pi}_{h+1:H}) - v(s_h; \hat{\pi}_{h:H}) \le \varepsilon_{s_h} \qquad (2)$$

with probability $1 - \delta/4S$. Now suppose (2) holds for every $h \in [H]$ and $s_h \in \mathcal{S}_h$, which holds with probability $1 - \delta$. We have

$$V^{\pi^*}(s_0) - V^{\hat{\pi}}(s_0) = \sum_{h=1}^{H} E_{s_h \sim P_h^*} \left[ v(s_h; \pi_h^* \circ \hat{\pi}_{h+1:H}) - v(s_h; \hat{\pi}_{h:H}) \right]$$

$$\leq \sum_{h=1}^{H} E_{s_h \sim P_h^*} \left[ v(s_h; \tilde{\pi}_h^* \circ \hat{\pi}_{h+1:H}) - v(s_h; \hat{\pi}_{h:H}) \right]$$

$$\leq \sum_{h=1}^{H} \Pr_{\pi^*}[s_h \notin \tilde{\mathcal{S}}] + \Pr_{\pi^*}[s_h \in \tilde{\mathcal{S}}] E_{s_h \sim P_h^*, s_h \in \tilde{\mathcal{S}}} \left[ v(s_h; \tilde{\pi}_h^* \circ \hat{\pi}_{h+1:H}) - v(s_h; \hat{\pi}_{h:H}) \right]$$

$$\leq \varepsilon/(2S) \cdot S + \sum_{h=1}^{H} \sum_{s_h \in \tilde{\mathcal{S}}_h} P_h^*(s_h) \left( v(s_h; \tilde{\pi}_h^* \circ \hat{\pi}_{h+1:H}) - v(s_h; \hat{\pi}_{h:H}) \right)$$

$$\leq \varepsilon/2 + \sum_{h=1}^{H} \sum_{s_h \in \tilde{\mathcal{S}}_h} P_h^*(s_h) \varepsilon_{s_h}.$$

Here the first equality is the performance different lemma (Lemma 13, [24]), and $P_h^*$ is the state distribution of $\pi^*$ on step $h$. The first inequality comes from the definition of $\tilde{\pi}^*$. Now note that $P_h^*(s_h) \leq \mu(s_h)$, and that $\sum_{h=1}^{H} \sum_{s_h \in \tilde{\mathcal{S}}_h} P_h^*(s_h) = H$; we have

$$\sum_{h=1}^{H} \sum_{s_h \in \tilde{\mathcal{S}}_h} P_h^*(s_h) \varepsilon_{s_h} = \frac{\varepsilon}{2S^p H^q} \sum_{h=1}^{H} \sum_{s_h \in \tilde{\mathcal{S}}_h} P_h^*(s_h) \mu^{-p}(s_h)$$

$$\leq \frac{\varepsilon}{2S^p H^q} \sum_{h=1}^{H} \sum_{s_h \in \tilde{\mathcal{S}}_h} (P_h^*(s_h))^{1-p} \leq \varepsilon/2.$$

The inequality holds from Hölder's inequality. Therefore we show that the output policy is $\varepsilon$-optimal.

Now we compute the sample complexity. The step complexity is simply $O(HSN_0)$. For comparison complexity, we roll out at most $SN_0$ trajectories, so the comparison complexity is at most $SN_0$.

$\square$

### D.6 Proof of Theorem 9

*Proof.* The proof of Theorem 9 is largely the same as Theorem 7. For every state $s$, let $\tilde{\mu}(s)$ be the probability that $\hat{\pi}_s$ visits $s$. Similar to Theorem 5 and 7, we have $N_s \geq 1/2\mu(s)N_0$ for $s \in \tilde{\mathcal{S}}$. Therefore by randomly pick a policy from the $N_0$ episodes, we have $\tilde{\mu}(s) \geq 1/2\mu(s)$.

Using Hoeffding's inequality and property of EULER we have that with probability $1 - \delta/2$, for every state $s$ we have

$$\hat{R}_{s_h}/N_1 \geq \tilde{\mu}(s) - \sqrt{\frac{\log(4S/\delta)}{N_0}} \geq 1/2\mu(s) - \sqrt{\frac{\log(4S/\delta)}{N_0}},$$

and therefore $\mu(s) \leq \hat{\mu}(s)$. On the other hand, we also have

$$\hat{\mu}(s) \leq 2\hat{R}_{s_h}/N_1 + 2\sqrt{\frac{\log(4S/\delta)}{N_0}} \leq 2\tilde{\mu}(s) + 4\sqrt{\frac{\log(4S/\delta)}{N_0}} \leq 2\mu(s) + 4\varepsilon/S \leq 6\mu(s).$$

Define $\varepsilon'_{s_h} \leftarrow \frac{\varepsilon}{2(\mu(s_h)SH^{\alpha-1})^{1/\alpha}}$ as in Theorem 7. Therefore we know that with probability $1 - \delta/2$, we have $\varepsilon_{s_h} \leq \varepsilon'_{s_h}$ and that $\varepsilon_{s_h} = \Theta(\varepsilon'_{s_h})$. The rest of the proof follows the same process as Theorem 7.

$\square$

# E Auxiliary Lemma

We present the performance difference lemma here for completeness. Here we use the version adapted to episode MDPs as in [24].

**Lemma 12** (Lemma 13, [24]). *For any episode MDP with reward function $r$ and two policies $\pi_{0:H-1}$ and $\pi'_{0:H-1}$, For any $h \in [H]$, let $Q_h(s)$ be the distribution of state $s$ at step $h$ induced by policy $\pi_{0:H-1}$. We have*

$$v_{\pi_{0:H-1}}(s_0) - v_{\pi'_{0:H-1}}(s_0) = \sum_{h=0}^{H-1} E_{s \sim Q_h(s)}[V_{\pi_h \circ \pi'_{h+1:H}}(s) - V_{\pi'_{h:H}}(s)].$$