[Reviews · NeurIPS 2020]

Review 1

Summary and Contributions: In the context of preferenced-based RL, this paper identifies an assumption on the answers to the preference queries and introduces an algorithm such that, when combined, this results in recovering net-optimal policies with respect to the underlying reward function generating the comparison feedback.

Strengths: The paper provides insight into preference-based RL. It shows that a winning policy might not actually exist if people were to give perfect trajectory comparisons, which is in itself an important point. It also shows that under the assumption of the probability of the answer being a linear function of the difference in cumulative reward, an algorithm exists that yields epsilon-optimal policies, studying the sample complexity in both comparisons required and exploration steps. Concretely, the results establish that the exploration complexity of learning from comparisons can be similar to that of value based RL in the simulator setup and slightly worse in the exploration setup.

Weaknesses: There are two main weaknesses. First, I’m not sure whether the algorithm is meant to be the core contribution, or the analysis. If it’s the algorithm, then the paper needs to actually test the algorithm in more than toy settings (and ideally with real humans, rather than simulating answers with BLT with two parameter settings). But if it’s the analysis, I almost feel like the experiments are distracting, or at least overstating and drawing away from the main contributions. I’d love to hear the authors’ perspective on this, but my suggestion would be to either a) get the best of both worlds by running a more serious experiment, or b) edit the paper to highlight the analysis and justify the experiments as showing what the algorithm does empirically and perhaps aiding with some qualitative analysis of the resulting behavior when applied to simple tasks, aiding in the understanding of the algorithm. Second, the assumption this is based on is very strong. It is pretty clear that real people won’t satisfy it, as decades of behavioral economics research has shown us. Two things would make the analysis much stronger: expanding it to a family of models, and providing an analysis of how when the assumption does not hold, how the approximation error would propagate into the bounds. For the former, a function like f(difference) instead of linear, including the BLT model, would be nice. Further potential weaknesses: - Prop 1 shows that a Condorcet winner amongst policies might not exist due to cyclic preferences induces by trajectory preferences. However, recent work by [1] showed that a von Neumann winner always exists. Using this as motivation for switching to the strong assumption 1 only makes sense if one can show that the von Neumann winner (which is a stochastic policy now) would have poor performance with respect to the value function in the underlying MDP. [1] Dudík M, Hofmann K, Schapire RE, Slivkins A, Zoghi M. Contextual dueling bandits. arXiv preprint arXiv:1502.06362. 2015 - Assumption on reward scaling: Lines 87-88 mention that both the rewards and cumulative value of H states are bounded between [0,1]. If the rewards are between [0,1] shouldn't the correct scaling of the values be from [0,H]? How does this affect the scaling of the sample complexity bounds? - Assumption on state space: in Line 81, the paper assumes that the state-space can be partitioned into H sets. What this implies is that if the same set of states can be encountered at each time-step, the effective size of the state space is S_eff = S H. I believe that the bounds in this case would incur additional H factors, making it sub-optimal? Is this assumption necessary? Is this also assumed in prior work and the bounds compared with for the value-based RL setup? If not, I believe that the comparisons with the existing bounds might be unfair {same for point d above}. More minor points on experiments: - For the synthetic experiments, the comparisons are generated according to a BTL model using the underlying reward function. How robust are the findings if we change this model? - Even for the BTL model, it would be good to have a plot with varying c for the proposed algorithm to get an understanding of the robustness properties of the procedure to noise in comparisons. - The experiments seem quite small scale with the synthetic grid-world (4x4) and Random MDP (5 steps, 4 states per step). How well does the overall method do on large scale setups? It would be good if the procedure could also be run on for instance, the mountain car (and other such envs) as was done in Novoseller et al. 2019. - Scaling plots: In order to understand if the scaling with the terms S, A and H are indeed tight, it would be good to have experiments which study these scaling laws.

Correctness: At a first glance, the proofs in the appendix look fine to me and the results stated in the main paper seem correct under the assumptions listed.

Clarity: Overall, the paper is well motivated and well-written. The theorem statements are precise and the paper does a good job of comparing the different results in the form of corollaries and discussions. Small nitpick: It’d be useful after assumption 1 to write one sentence that now a Condorcet winner exists, because preferences are based on value and there exists a value maximizing policy. This would just help exposition, of course it's implicit right now. There are a couple of minor typos which need to be addressed: a. In the MDP formulation, it looks like the MDP is deterministic and so are the policies but the paper mentions randomness in the MDP (see Prop 1 and thereon). I believe that the transition model needs to encode random transitions. b. Definition 1: C(\eps, \delta) -> \Psi(\eps, \delta) c. Proposition 2: C -> C' d. Corollary 4: Should be log(S/\delta) e. Proof of prop 1: should be pi0, pi1, pi2 (rather than pi1, pi2, pi3)

Relation to Prior Work: The paper compares positions itself with relevant prior work in the first couple of pages and also compares the established results with previous known results in both preference based and value based reinforcement learning. It would be nice if for completeness, some insights into the dueling bandits framework could be provided in the appendix for readers not familiar with the topic.

Reproducibility: Yes

Additional Feedback:


Review 2

Summary and Contributions: This work provides contributions to the preference based reinforcement learning (PbRL) problem. Contrary to classical RL where numerical rewards are observed, only comparisons between trajectories are obsersved in PbRL. Intuitions about this problem are given by discussing what happens under different assumptions in Propositions 1 and 2. Then, algorithms using RL and dueling bandit algorithms as subroutines, are described and analysed in terms of both step and comparison complexities required to find the optimal policy in a PAC setting. Finally, numerical experiments are provided as illustrations of the theory.

Strengths: This paper provides fundamental contributions to the PbRL problem, which is by far less well studied and understood than the RL problem, in spite of its potential broad application in contexts where numerical rewards are missing.

Weaknesses: Some assumptions may be too constraining for real world applications.

Correctness: The method is correct. The empirical methodology corresponds to the theoretical setting.

Clarity: The paper is very well written.

Relation to Prior Work: Yes, related PbRL works are recalled, as well as dueling bandits algorithms that are used as a subroutine.

Reproducibility: Yes

Additional Feedback: Some typos: - line 76: two sentences should be permuted - line 79: did you mean "random" instead of "randomized"? - line 110: C(eps, delta) should be replaced by Psi(eps, delta) - line 112: precision "for any arm a different from hat{a}" - lines 137 and 139: I think that s_h should be replaced by s - lines 157-158: choose either C or C'


Review 3

Summary and Contributions: The paper looks at preference-based reinforcement learning from a sample complexity standpoint. In preference-based reinforcement learning, feedback is provided on pairs of trajectories to say which is better/preferred. Thus, the feedback is weaker than traditional RL in two respects. First, comparisons are qualitative (x > y) and not quantitative (x is worth R(x)). Second, information about the utility of individual steps in the trajectory is lost and comparisons are made in the aggregate.

Strengths: The analysis seems thorough and novel.

Weaknesses: This paper does not make an explicit case for the utility of the preference-based framework. It seems interesting and arguably understudied. But, I think more of an argument should be made for its significance. Alternatively, I'd find it interesting to learn more about the technical details of developing algorithms for this class of problems. Instead, a lot of the technical meat seems to be hidden in the behavior of dueling bandit algorithms.

Correctness: The analysis seems appropriate and correct.

Clarity: Paper is mostly clear, but I would have felt better informed to hear more about the internals of preference-based RL algorithms / dueling bandits. To me, the paper read as if it were an addendum to a prior paper and not really a standalone presentation.

Relation to Prior Work: Yes, relation to both earlier preference-based RL work and sampling complexity analyses of the classical RL setting is addressed.

Reproducibility: No

Additional Feedback: Post author response: I thank the authors for your response. I had been uninformed about preference-based RL and so I began to read the survey paper you shared. I still have misgivings about the paper, but I acknowledge that another of the reviewers (as well as the area chair) have expertise more directly relevant to the paper and I am comfortable giving their reviews/recommendations more weight. Detailed comments: "Although lower bounds for these bandit problems extends to PbRL," -> "Although lower bounds for these bandit problems extend to PbRL,"? "For simplicity we assume S >=H": "S" hasn't been defined yet. Maybe move the sentence over by one? Proposition 1 is interesting. I have played with these kinds of non-transitive distributions. I suspect some readers may be unfamiliar with the phenomenon. Maybe include a citation? Even a wikipedia reference might be useful (https://en.wikipedia.org/wiki/Nontransitive_dice). "a MDP" -> "an MDP"? "In this Section," -> "In this section," "Preferece-based Exploration" -> "Preference-based Exploration" "reward-free learning literatures" -> "reward-free learning projects"? "we compare them and feed it back": What is "it" referring to here? "PEPS almost always get" -> "PEPS almost always gets"


Review 4

Summary and Contributions: This paper presents a finite time analysis of preference-based reinforcement learning. Overall, it fills up a blank piece of preference-based RL. Moreover, a new algorithm is proposed that has a theoretical guarantee.

Strengths: The theoretical analysis is a significant contribution.

Weaknesses: The assumption might be overly strong and unrealistic. There are some errors in the proof. Need to be rechecked.

Correctness: there are some issues in the proof (line number refers to the full paper file) in line 493, the second inequality, s_h \in \tilde S should be s_h \notin \tilde S. in line 508, the same issue in line 511, = H should be \leq H please recheck the proofs

Clarity: The paper is easy to follow.

Relation to Prior Work: The difference is clear.

Reproducibility: Yes

Additional Feedback: An overall major concern of this paper is whether Assumption 1 is overly strong. Note that when a practitioner gives a reward function, he/er is commonly not fully aware of the total reward, i.e., the V value. This happens very often, as practitioners need to adjust their reward functions for a good performance. What Assumption 1 says is that, when the preference is given, it will not lose good policies. If a practitioner is able to do that, he/er is likely to give a good reward function. So can the assumption be release to a probabilistic inequality, e.g., φ_s (\pi_1,\pi_2) ≥ C(v_{\pi_1}, v_{\pi_2}) with a probability? How will this change affect the analysis?

[Author Response · NeurIPS 2020]



(a) Linear preference model, Random MDP  (b) Deterministic preference, GridWorld  (c) Varying $c$ in BTL model, GridWorld  (d) Varying $H$, GridWorld

We thank the reviewers for their thoughtful comments and we respond to the major questions below.

**Reviewer 1**  *Q1: Core contribution of the paper.* Our core contribution is to understand the theoretical rate of PbRL, and how they
are different from traditional value-based RL. Our experiments do not aim to show strong performance in real applications; instead,
we find that existing PbRL algorithms can suffer from long convergence time in estimating the value function. We hope our method
can inspire better performing PbRL algorithms.

*Q2: The assumption is very strong.* If we replace the $C_0(v(\pi_1) - v(\pi_2))$ in Assumption 1 by $f(v(\pi_1) - v(\pi_2))$ for some function
$f$ (e.g., logistic for BTL model), then our result still holds as long as $f$ is Lipschitz lower bounded; so our methods work for the
BTL model. On the other hand, if we impose a BTL model on trajectory preferences, one cannot recover the correct optimal policy
similar to the deterministic preference case. For example, suppose $\pi_1$ gets reward $0.5 + \varepsilon$ with probability 1 for some $\varepsilon > 0$, and $\pi_2$
gets reward 0.75 with probability 2/3 and 0 otherwise. With some calculation one can show that $\pi_1$ loses to $\pi_2$ with probability
larger than 0.5 for $\varepsilon = 0.001$. Actually, if we multiply all the rewards in Proposition 1 by a large constant, BTL will become close to
deterministic and the proposition will hold for BTL. We conjecture that Proposition 1 will be true for *any* non-linear function $f$ under
the $f$(difference) comparisons model. Under deterministic transitions, our Assumption 1 holds for a large family of comparison
models including BTL and deterministic. It is interesting future work to check the quality of the recovered policy when Assumption
1 holds with some error.

*Q3: von Neumann winner.* This is a very nice suggestion. However, von Neumann winner requires a distribution of policies whose
support is the on the whole policy space. The policy space is exponentially large so it can be exponentially hard to recover the von
Neumann policy. But we agree this would be an interesting avenue for future work.

*Q4: Assumption on reward scaling and state space.* We need an extra assumption that the total reward is between $[0, 1]$ so that $c$ in
Assumption 1 can be a constant. If we instead assume all rewards are in $[0, 1/H]$, the step and comparison complexities will be less
by a factor of $O(H^2)$. Traditional value-based RL literature has considered this setting as well, see references 20 and 26 and Line
119-126 in our paper. The disjoint state space assumption is common in prior works, e.g., reference 14 and 23 in the paper. So our
results can be compared fairly with previous work. We will make this point clear in our final version.

*Questions on experiments.* We have performed extra experiments, and we provide some examples above. We have tested the linear
comparison model and deterministic (exact) comparisons (figure a,b), and tested the effect of $c$ in BTL model (figure c). We focus on
small-scale experiments as our goal is only to illustrate the ideas, similar to prior work (e.g., reference 14, 23 in paper). In Figure (d)
we test the regret versus the time horizon $H$. It shows a close to linear relation, which fits our rate in Corollary 8 (the $O(H^2/\varepsilon^2)$
term translates to a linear dependence of $\varepsilon$ on $H$). We will include plots verifying scaling with $S, A, H$ in our final version.

**Reviewer 2** *Q: Some assumptions might be too constraining.* Our assumptions are necessary to ensure that the true optimal policy
can be recovered from the preferences (see Q2, Reviewer 1).

**Reviewer 3** *Q1: Significance of the PbRL framework.* PbRL is widely applied in previous research to combat problems like reward
hacking and help with reward engineering, and we refer the reviewer to reference 27 in our paper for an overview. By replacing
numerical rewards with human preferences, PbRL not only reduces the effort in reward engineering but also in reward shaping,
where the rewards help the agent to find the optimal policy. PbRL has a wide application in robot training [1] and game playing
(reference 11,27 in the paper). As we stated in the introduction, there is NO existing work with a finite-time guarantee to the best of
knowledge, and we propose the first PbRL algorithm with guaranteed performance. Our results (Proposition 1,2, Assumption 1) also
establish the necessary conditions on preference probabilities to make sure that the optimal policy is recoverable.

*Q2: Technical Details.* Our technical contribution is mainly two-fold. Firstly, we characterize the conditions on the preference
probabilities to recover the optimal policy. Different than dueling bandits, the deterministic or BTL model (see Q2 of Reviewer 1
above) does not work for PbRL. Secondly, we show a reward-free way to guide the exploration (our PEPS algorithm) when we do
not have access to the reward values in each step. We cannot compute the value function in PbRL because reward values are hidden.
We use a synthetic reward function (see Sec 4.1) to guide the exploration of PbRL. While our algorithm is based on existing results
in dueling bandits, developing algorithms for PbRL is much harder and dueling bandits is just a building block.

**Reviewer 4** *Q1: The assumptions are overly strong.* We believe that the reviewer has a misunderstanding of our assumption. Our
definition of $\phi_s(\pi_1, \pi_2)$ (see first line of Proposition 1) is defined as the probability that a random *trajectory* from $\pi_1$ beats a random
trajectory from $\pi_2$; it already **includes** the randomness in the transitions and preference probability. This does not mean that a
good policy will never lose to a worse policy, and also it does not have to win under all trajectories; we only need to assume that
it wins with a large probability under the distribution of trajectories. Our Assumption 1 states the exact point that a trajectory $\tau_1$
from $\pi_1$ only beats a trajectory $\tau_2$ from $\pi_2$ with a probability, and we assume that the overall probability of $\tau_1$ beating $\tau_2$ is at least
$C_0(v(\pi_1) - v(\pi_2))$, over the random draws of the trajectories. We do not make assumptions on individual trajectories and our
assumption is a relatively mild one. Moreover, we have shown that more traditional assumptions like deterministic and BTL (see
Proposition 1 and Q2 in reviewer 1) cannot correctly recover the optimal policy.

The errors that the reviewer points out are typos that we will correct in our final version. On line 511, $P_h^*$ is defined as the state
distribution of $\pi^*$, so the equation should be exact "=".

**References**  [1] Kupcsik, Andras, David Hsu, and Wee Sun Lee. "Learning dynamic robot-to-human object handover from human
feedback." Robotics research. Springer, Cham, 2018. 161-176.

[Meta-Review · NeurIPS 2020]

This paper generated considerable discussion among the reviewers. One the positive side, this paper makes a solid contribution to the emerging literature on preference-based RL, a topic of some importance and makes some interesting insights (e.g., on the potential lack of a “winning policy”) and novel algorithmic contributions. Conversely, some reviewers raised issues with some of the assumptions made in the paper and the presentation (which seems to assume familiarity with PBRL and its motivations/rationale. The author response was thoughtful and generated some discussion (some of which is not reflected in the reviews, a couple of which failed to get updated unfortunately). On my own reading if the paper, I agree that the paper makes a useful contribution to PBRL, especially from a technical perspective and conceptual perspective (although I don’t believe it makes PBRL more practical at this stage). Apart from some weaknesses raised in the other reviews, and some limitations (which could lead to extensions in future work), I do have some methodological qualms with the paper. Specifically, the direct application of dueling bandit methods over trajectories to drive comparisons of policies seems methodologically flawed. It is “straightforward” humans/users compare trajectories (noise-free or noisily). But to effectively translate this into a "vote" for the value of a policy that (could have) induced it seems to be simply a way of trying to directly apply dueling bandits without much additional work. IMO, the reason we ask users to compare trajectories is to get a sense of the reward function. The induced (possibly stochastic) constraints on the reward function can then be directly translated into constraints on the value function. I don’t see a legitimate rationale for wanting to think of these as stochastic samples of "policy comparisons". From that perspective, Prop 1 seems much less interesting than the paper claims, and is, if not a "formal" consequence, then at least an "intuitive" consequence of social choice/preference aggregation theory. However, this type of approach is a part of some important PBRL frameworks, and as such, I believe it is worthy contribution to the PBRL literature, and the scientific community should judge the value/legitimacy of some of the assumptions made. Hence, I recommend acceptance. The reviewers make many detailed and valuable suggestions, and the author(s) is (are) strongly encouraged to revise the paper to account for these.